# Possible Involvement of 2′,3′-Cyclic Nucleotide-3′-Phosphodiesterase in the Protein Phosphorylation-Mediated Regulation of the Permeability Transition Pore

**DOI:** 10.3390/ijms19113499

**Published:** 2018-11-07

**Authors:** Yulia Baburina, Irina Odinokova, Tamara Azarashvili, Vladimir Akatov, Linda Sotnikova, Olga Krestinina

**Affiliations:** Institute of Theoretical and Experimental Biophysics, Russian Academy of Sciences, Pushchino 142290, Russia; byul@rambler.ru (Y.B.); odinokova@rambler.ru (I.O.); tazarash@rambler.ru (T.A.); akatov.vladimir@gmail.com (V.A.); linda_sotnikova@mail.ru (L.S.)

**Keywords:** permeability transition pore, protein phosphorylation, serine/threonine kinases, calmodulin, 2′,3′-cyclic nucleotide 3′-phosphodiesterase, melatonin

## Abstract

Calcium as a secondary messenger regulates the phosphorylation of several membrane-bound proteins in brain and liver mitochondria. Regulation of the activity of different protein kinases and phosphatases by Ca^2+^ occurs through its binding with calmodulin. The protein phosphorylation is strongly dependent on the Ca^2+^-induced mitochondrial permeability transition pore (mPTP) opening. 2′,3′-Cyclic nucleotide-3′-phosphodiesterase (CNPase) was phosphorylated by protein kinases A and C. CNPase and melatonin (MEL) might interact with calmodulin. The effects of the calmodulin antagonist calmidazolium and the inhibitor of protein kinase A H89 on mPTP opening in rat brain mitochondria of male Wistar rats were investigated. In addition, the role of CNPase, serine/threonine kinases, and MEL in the mPTP opening was examined. The anti-CNPase antibody added to rat brain mitochondria (RBM) reduced the content of CNPase in mitochondria. The threshold [Ca^2+^] decreased, and mitochondrial swelling was accelerated in the presence of the anti-CNPase antibody. H89 enhanced the effect of anti-CNPase antibody and accelerated the swelling of mitochondria, while CmZ abolished the effect of anti-CNPase antibody under mPTP opening. The levels of phospho-Akt and phospho-GSK3β increased, while the MEL content did not change. It can be assumed that CNPase may be involved in the regulation of these kinases, which in turn plays an important role in mPTP functioning.

## 1. Introduction

The protein phosphorylation/dephosphorylation system regulates many cell functions and is involved in signal transduction. It can significantly influence the structural and morphological features of protein [1]. Many kinases and phosphatases responsible for phosphorylation and dephosphorylation, respectively, accomplish their functions in the cell primarily outside mitochondria but it is known that many protein kinases, phosphatases, and related proteins are also contained inside mitochondria [1]. One of the main protein kinases found in mitochondria is serine/threonine-activated kinase Akt, also known as protein kinase B [2]. Several studies showed that mitochondrial Akt catalyzes a phosphorylation and inactivates the pro-apoptotic protein BAD, and the recruitment of Raf-1 to the mitochondria promotes cell survival [3,4]. Recent investigations discovered new targets for mitochondrial Akt such as Complex V of the mitochondrial electron transport chain and hexokinase-II [5,6]. Akt also has a direct effect in mitochondria, which is mediated by phosphorylation of hexokinase-II, resulting in the protection of mitochondria from oxidants or the opening of the Ca^2+^-induced mitochondrial permeability transition pore (mPTP) [6]. The mPTP opening results in the inhibition of respiration, mitochondrial swelling, the inner membrane depolarization, and the release of pro-apoptotic factors [7]. Another serine/threonine kinase, GSK3β, is present in mitochondria of the rat cerebellum [8]. GSK3β regulates the function of mitochondrial proteins such as adenine nucleotide translocator (ANT) and cyclophilin D [9,10,11]. It is known that the inhibition of GSK leads to a lag in the opening of mPTP in response to oxygen radicals [12]. It has been suggested that the mPTP opening is a causative event in cell death during myocardial ischemia/reperfusion impairment [13,14]. GSK3β is localized in the cytosol and the nucleus but its action focuses on mitochondrial proteins; therefore, the knowledge of its function is of great interest. It was reported that the content of total and phosphorylated GSK3β in mitochondria of isolated rat hearts subjected to ischemia/reperfusion increases, and the protein–protein interaction between GSK3β and VDAC or ANT enhances [15]. Phosphorylated GSK3β inhibits mPTP opening by a variety of mechanisms [16,17]. Ca^2+^, as a second messenger, might regulate the activity of different protein kinases and protein phosphatases through binding with calmodulin (CaM) [18]. CaM is a Ca^2+^-binding protein, which interacts with a great number of target proteins and regulates their functions. CaM is also found in great amounts in the nervous system where it interacts most intensively with the myelin basic protein, which has been confirmed by structural studies [19,20,21,22].

Recently a neuroprotective protein has been identified as 2′,3′-cyclic nucleotide-3′-phosphodiesterase (CNPase) in rat brain mitochondria (RBM), and it has been shown that CNPase protects mitochondria from mPTP opening [23]. It is known that CNPase (CNP, EC 3.1.4.37) catalyzes the hydrolysis of 2′,3′-cyclic nucleotides to form the corresponding nucleoside 2′-monophosphates [24]. The functions of CNPase in mitochondria are presently poorly understood. CNPase substrates (2′,3′-cAMP and 2′,3′-cNADP) were found to be able to accelerate mPTP opening in RBM as well as in rat liver mitochondria (RLM) [23], which indicates potential interaction of CNPase with the modulators of mPTP in mitochondria. CNPase interacts with RNA, CaM, and the cytoskeleton proteins [25]. It also inhibits translation to modulate mitochondrial membrane permeability and has putative ATP/GTPase activity. The addition of the anti-CNPase antibody to a mitochondrial suspension was found to protect mitochondria from mPTP opening [26,27]. CNPase is phosphorylated by protein kinase A (PKA) and protein kinase C [28].

The hormone of the pineal gland melatonin (*N*-acetyl-5-methoxytryptamine, MEL) is a highly conserved molecule found in a variety of organisms, from unicellular organisms to vertebrates [29,30]. Two isoforms of the melatonin receptor, MT1 and MT2, were found in the brain and peripheral tissues of mammals [31,32,33]. Both isoforms are expressed in brain mitochondria. It is believed that the interactions of MEL with its receptors may be involved in mitochondrial cell death, although melatonin (MEL) may also have other, non-receptor-mediated targets. MT1 was reported to be contained predominantly in the mitochondrial fraction [34,35]. Recently, we have found that MEL retains СNPase inside mitochondria to protect cells against the deleterious effect of 2′,3′-cAMP in aging [36]. Myllykoski and coworkers have shown that CNPase directly interacts with CaM in a calcium-dependent manner [37]. MEL interacts with CaM, G-proteins, protein kinase A, and adenylyl cyclase, which are involved in calcium signaling and the cAMP-signaling cascade [38]. The aim of the present work was to study the effect of the CaM antagonist calmidozolium (CmZ) and the PKA inhibitor H89 on the functional state of RBM upon mPTP opening and the role of serine/threonine kinases, CNPase, and MEL in this process.

## 2. Results

We measured the parameters of mPTP function such as Ca^2+^ retention capacity, oxygen consumption, and RBM swelling and compared them in different experimental conditions (in the presence of CmZ (30 μM), H89 (5 μM), the anti-CNPase antibody (0.18 µg/mL) and their combination).

At first, we examined the influence of these agents on the Ca^2+^ retention capacity of RBM upon mPTP opening (Figure 1). Under all experimental conditions, the first addition of Ca^2+^ (50 nmol/mg of protein) to RBM led to an intensive accumulation of Ca^2+^ in mitochondria (Figure 1a, curve 1). However, in control conditions, the sixth addition of Ca^2+^ (each addition after the first, 100 nmol of Ca^2+^ per mg of protein) to RBM was not loaded and led to the mPTP opening. The release of accumulated Ca^2+^ (mPTP opening) occurred after the sixth and fifth addition of Ca^2+^ to RBM in the presence of CmZ (curve 2) and H89 (curve 3), respectively. The anti-CNPase antibody induced mPTP opening after the fifth pulse (curve 4). The addition of CmZ and H89 to Anti-CNPase antibody-treated RBM led to Ca^2+^ release after the sixth and fifth pulses, respectively (curves 5 and 6). Figure 1b demonstrates quantitative changes in the Ca^2+^ retention capacity of Ca^2+^-loaded RBM under different conditions (Figure 1a). The threshold Ca^2+^ concentration (a calcium concentration that leads to the mPTP opening) was approximately 450 µM (Figure 1b, column 1). In the presence of Cmz, the threshold Ca^2+^ concentration in RBM increased by 10% (Figure 1a, curve 2, Figure 1b, column 2), while H89 (Figure 1a, curve 3) did not change the threshold Ca^2+^ concentration compared to the control (Figure 2b, column 3 vs. 1). Recently we have identified the neuroprotective protein CNPase in RBM and showed that CNPase protects mitochondria from mPTP opening [23]. Since, to our knowledge, there is no specific inhibitor for CNPase, we tested whether our monoclonal anti-CNPase antibody [39] is able to modulate CNPase-mediated effects on the functional parameters of mitochondria. We found that the threshold Ca^2+^ concentration of RBM in the presence of the monoclonal anti-CNPase antibody (Figure 1a, curve 4) was reduced by approximately 14% and was 390 µM (Figure 2b, column 4). The threshold Ca^2+^ concentration of RBM did not change by the combined action of CmZ and the anti-CNPase antibody (Figure 1a, comparison of curves 2 and 5; Figure 1b, columns 2 and 5). H89 strengthened the inductive effect of the anti-CNPase antibody under their combined action when mPTP was opened (Figure 1a, comparison of curves 3 and 6; Figure 1b, columns 3 and 6).

Under the same conditions, the respiratory activity of the RBM was measured (Figure 2). The experiments were carried out as described in Materials and Methods. Figure 2a shows the curves of mitochondrial respiration in experimental conditions. Figure 2b shows a schema for the calculation of oxygen consumption rates in state 2 (V^O2^_Ca_ St.2; ng-atom O min^−1^ mg^−1^ of protein), state 3 (V^O2^_Ca_ St.3; ng-atom O min^−1^ mg^−1^ of protein), and state 4 (V^O2^_Ca_ St.4; ng-atom O min^−1^ mg^−1^ protein). The evaluation of oxygen consumption rates in different states is represented in Figure 2c. As seen from the figure, no changes occur in the rate of substrate-dependent respiration (state 2) in all experimental conditions used. The addition of Ca^2+^ (the first pulse) led to an increase in the respiration rate in state 3 by 7% in the presence of CmZ and a decrease in the respiration rate by 7% and 20% in the presence of anti-CNPase and anti-CNPase combined with CmZ, respectively. In the presence of CmZ, the rate of oxygen consumption (State 4) in RBM was suppressed by 30% relative to the control. The anti-CNPase antibody did not change the respiratory rate of RBM in state 4 (V^O2^_Ca_ St.4) in comparison with control, while the combined effect of anti-CNPase antibody and CmZ decreased the respiration rate (V^O2^_Ca_ St.4) by 15%. H89 also did not affect the activation of oxygen consumption in RBM (V^O2^_St.4_), but the combined action of the anti-CNPase antibody and H89 suppressed the oxygen consumption in RBM (V^O2^_Ca_ St.4) by 15% compared to the control.

The addition of Ca^2+^ at the threshold concentration to the mitochondrial suspension incubated in the standard medium described in Materials and Methods caused a decrease in light scattering, which is indicative of mitochondrial swelling. At the next step, we compared the swelling of RBM in different experimental conditions. Figure 3a shows the curves of Ca^2+^-activated swelling of RBM. In Figure 3b, the average half-time of mitochondrial Ca^2+^-activated swelling (T_1/2_) is given. In the presence of CmZ and H89, the half-time of mitochondrial swelling increased by 45 and 37%, respectively. Thus, the rate of swelling of RBM decelerated as compared to the mitochondrial swelling in control conditions (without additions), while the time needed for the swelling of RBM after the addition of the anti-CNPase antibody decreased by 30%. The anti-CNPase antibody tended to accelerate the swelling of RBM compared with the swelling in the control. The combined action of the anti-CNPase antibody and CmZ abolished the effect of anti-CNPase antibody used alone. RBM swelling slowed down two times compared with the control (anti-CNPase antibody). The combined action of the anti-CNPase antibody and H89 abolished the effect of the anti-CNPase antibody. RBM swelling decreased by 42% compared with the control (anti-CNPase antibody alone).

The activation of pro-survival protein kinases (Akt, GSK3β, and others) has been implicated as one of the signaling pathways. These pathways alter the mPTP that opens in response to oxidative stress and is responsible for the induction of programmed cell death [14]. Here we analyzed alterations in the ratio of the phospho-GSK3β-to-total GSK3β (pGSK3β/GSK3β) (Figure 4a) and the ratio of the phospho-Akt-to-total Akt (pAkt/Akt) (Figure 4b) in different experimental conditions in RBM. The pGSK3β/GSK3β ratio (Figure 4a) increased two times in the conditions when mPTP was opened (column 2 vs. 1).

This ratio increased by 60% when CmZ at the threshold Ca^2+^ loading was added (column 3 vs. 2) and did not change after the addition of H89 at the threshold Ca^2+^ loading compared with that in the case of Ca^2+^-loaded mitochondria (column 7 vs. 2). In the presence of the anti-CNPase antibody, the pGSK3β/GSK3β ratio increased 1.75 times (Figure 4a, column 4 vs. 1). Under the combined action of the anti-CNPase antibody and Ca^2+^ loading (the mPTP is opened; Figure 4a, column 5 vs. 4), the pGSK3β/GSK3β ratio rose by 17%. However, after the addition of CmZ (column 6 vs. 5) and H89 (column 8 vs. 5) in combination with the anti-CNPase antibody, the pGSK3β/GSK3β ratio diminished by 16 and 35%, respectively.

The pAkt/Akt ratio (Figure 4b) did not change when mPTP was opened (column 2 vs. 1). The addition of CmZ increased the pAkt/Akt ratio by 27% (column 3 vs. 2), and in the presence of H89 it did not significantly change (column 7 vs. 2). In the presence of the anti-CNPase antibody, the pAkt/Akt ratio (column 4 vs. 1) increased 4.5 times relative to the appropriate control.

On the other hand, the pAkt/Akt ratio was reduced 1.85 times in the presence of the anti-CNPase antibody and Ca^2+^ loading (when mPTP was opened; column 5 vs. 4). When CmZ and H89 were used in combination with the anti-CNPase antibody, no significant changes in the Akt/Akt ratio were observed (columns 6 and 8 vs. 5).

In our previous studies, we assumed that CNPase was capable of protecting cells [40], and MEL is involved in the preservation of CNPase within mitochondria [36]. Here, we determined the contents of CNPase and melatonin receptor A1 (MT1) in RBM in our experimental conditions (Figure 5). The Western blot data presented in Figure 5 (upper part) show alterations in the MT1 and CNPase levels in RBM in different conditions. The results of the quantitative analysis of MT1 and CNPase levels are shown in Figure 5 (lower part). The intensity of protein bands was quantified after normalization with respect to COX IV.

In the presence of the threshold Ca^2+^ concentration, the content of MT1 decreased by 25%, while the content of CNPase did not change (columns 2 vs. 1). In the presence of CmZ upon mPTP opening, the content of CNPase diminished by 22%, whereas the content of MT1 did not change (columns 3 vs. 2). Conversely, in the presence of H89 when mPTP was opened, the content of CNPase did not change and the MT1 level decreased by 46% (columns 7 vs. 2).

The addition of the anti-CNPase antibody to RBM when the mPTP was closed did not influence the MEL and CNPase levels (columns 4 vs. 1). The addition of Ca^2+^ at the threshold concentration to anti-CNPase antibody-loaded mitochondria did not change the content of CNPase (columns 5 vs. 4), but the level of MEL increased by 30% compared with that in Ca^2+^-loaded mitochondria in the absence of the anti-CNPase antibody (columns 5 vs. 2). However, the level of CNPase increased by the combined action of CmZ and the anti-CNPase antibody (when mPTP was opened) by 30% compared with that in the presence of the anti-CNPase antibody alone under mPTP opening (column 6 vs. 5). In these conditions, the level of MEL did not change. We noticed a decrease in the MEL level by 60% and an increase in the CNPase level by 80% under the combined effect of H89 and the anti-CNPase antibody (when mPTP was opened, columns 8 vs. 5).

## 3. Discussion

CNPase is one of the most abundant proteins in the central nervous system [41]. Recently, we have shown that the protein identified as CNPase is localized in both mitoplasts and the outer mitochondrial membrane of rat brain and liver mitochondria and supposed that the enzyme participates in the regulation of mPTP opening. 2′,3′-cAMP and 2′,3′-cNADP (CNPase substrates) activated mPTP in RBM and RLM [23]. Lappe-Siefke and coworkers studied the role of CNPase at the translation level in knocked mice and found that CNPase causes the axonal swelling, which leads to the premature death [42]. It was proposed that CNPase may be a partner for CaM [43]. CaM has different target proteins in the brain [44,45]; thus, it is important for Ca^2+^-dependent phosphorylation of proteins in synaptosomal membranes [46]. In the middle of the 1980s, Hatase et al. detected CaM in mitochondria by immuno-electron microscopy. Using the complexes of anti-calmodulin antibody and protein A-gold, the authors showed that CaM in ultra-pure mitochondria is localized on the inner mitochondrial membrane and in the matrix space [47]. CNPase can interact with membrane surfaces, cytoskeletal proteins, and CaM [25]. CNPase was shown to facilitate microtubule polymerization and reorganization [48], and it can be hypothesized that CaM binding modulates this effect, depending on the intracellular calcium concentration [37]. CmZ is a general CaM antagonist, which prevents CaM from mediating calcium signaling [49]. Here we demonstrated the effect of CmZ and anti-CNPase antibody on mPTP opening. We found that CmZ increased the threshold Ca^2+^ concentration and slowed down the mitochondrial swelling, whereas the anti-CNPase antibody, while binding to the protein, decreased the threshold Ca^2+^ concentration, accelerated mitochondrial swelling and induced mPTP opening. Evidence obtained earlier indicated that CNPase is localized in the outer membrane and the matrix of RBM [23]. Presumably, in our case the anti-CNPase antibody is able to bind with CNPase on the outer mitochondrial membrane and eliminate its protective action of CNPase on mPTP opening. We supposed that the effect of CmZ on mitochondria might be due to its positive charge; this makes likely its accumulation by coupled mitochondria that have a high negative ΔΨ_m_, on the matrix side of the inner membrane. Recently we have shown that ^32^P-incorporation into the 17 kDa protein and into the 3.5 kDa polypeptide was enhanced in the presence of CmZ [50]. We found that CaM-dependent enzymes are involved in phosphorylation/dephosphorylation of these proteins [50]. In the present work, we showed that CmZ abolishes the stimulatory effect of the anti-CNPase antibody on mPTP opening in RBM. Presumably, CmZ leads to the interruption of CNPase–CaM interaction.

Earlier, we have found CNPase to be phosphorylated by cAMP-dependent PKA in RBM [28]. Here, the effect of the PKA inhibitor (H89) and the combined effect of H89 with the anti-CNPase antibody were investigated. We observed that H89 strengthened the effect of anti-CNPase antibody. The effect of H89 on sensitivity to mPTP opening by the anti-CNPase antibody was consistent with our recent findings that a decrease in CNPase expression leads to the induction of mPTP opening [23].

Several studies showed that the functioning of mitochondrial proteins such as adenine nucleotide translocator (ANT) and cyclophilin D are regulated by GSK3β [9,11,51]. In mitochondria, a key mitochondrial enzyme, pyruvate dehydrogenase, is phosphorylated and suppressed by GSK3β [52]. The role of GSK3β in determining the threshold for mPTP opening was demonstrated [12,53]. The pGSK3β/GSK3β ratio correlates with the threshold Ca^2+^ concentration needed to induce mPTP opening in mitochondria isolated from rat cardiomyocytes [54]. In our conditions, CmZ retarded mPTP opening, which led to the upregulation of the pGSK3β/GSK3β ratio. The addition of CmZ to anti-CNPase antibody-loaded RBM abolished the acceleration of mPTP opening caused by the anti-CNPase antibody and negligibly decreased the pGSK3β/GSK3β ratio. The addition of H89 to anti-CNPase antibody-loaded mitochondria even more enhanced the mPTP opening and notably diminished the pGSK3β/GSK3β ratio. Besides, phospho-inactivation of the pro-apoptotic protein BAD in mitochondria and the recruitment of Raf-1 to the mitochondria, which promoted cell survival, were caused by another serine/threonine kinase, Akt [3,4]. Recently, new targets for mitochondrial Akt have been discovered, in particular complex V of mitochondrial respiratory chain and hexokinase-II [5,6]. CNPase is associated with all respiratory chain complexes [55]. Moreover, structure of CNPase contains the Walker A motif, which is involved in the ATP binding, and the Walker B motif, which is able to coordinate Mg^2+^ ion that drives ATP hydrolysis [56,57]. We noticed that CmZ increased the pAkt/Akt ratio in RBM when mPTP was opened, and the addition of CmZ to the anti-CNPase antibody-loaded RBM abolished the effect. In the absence of anti-CNPase antibody we observed of correlation between pAkt/Akt and pGSK3β/GSK3β ratios. On the other hand, correlation between pAkt/Akt and pGSK3β/GSK3β ratios was broken when anti-CNPase antibody was added to mitochondria suspension. We suggests that CNPase may be involved in the regulation of these kinases. Recently we have shown that CNPase and MEL are interrelated in liver mitochondria isolated from old rats and MEL retained CNPase inside mitochondria to protect cells against deleterious effects [36]. Besides, we observed that the expression of CNPase did not change in RBM when mPTP was opened, whereas its activity strongly decreased [23]. The data obtained in this study indicate that the antibody specific to CNPase, did not decrease the CNPase level in phosphorylated samples of RBM when mPTP opened/closed. However, after the addition of CmZ and H89 to anti-CNPase antibody-treated mitochondria, the content of CNPase (when mPTP was opened) increased compared with that after the addition of the antibody alone. CmZ did not influence MEL receptor level but the addition of H89 to RBM led to a decrease in the level of MEL. After addition of H89 to anti-CNPase antibody-loaded RBM resulted in a more intensive increase in the MEL content.

It is known that MEL cooperates with CaM, adenylyl cyclase, protein kinase, and G-proteins which are participated in calcium signaling and the cAMP-signaling cascade [58]. CNPase has a consensus sequence of G-proteins [59], participates in the adenosine pathway [60,61], and is involved in the hydrolysis of toxic 2′,3′-cAMP in a CaM-dependent manner [62]. Gravel et al. suggested that the activation of PKA is necessary for the induction of the CNP1 mRNA accumulation by dbcAMP in glioma C6 cells [63]. We showed that CNPase is phosphorylated by PKA [28]. Thus, our data suggest that CNPase might participate in a complicated signal transduction system in mitochondria.

In summary (Figure 6), the addition of the anti-CNPase antibody led to a decrease in the threshold [Ca^2+^] decreased, and mitochondrial swelling was accelerated compared to the control, which agrees with the earlier made assumption that CNPase is involved in mPTP regulation [23]. The level of CNPase in the presence of the anti-CNPase antibody did not change, probably due to the binding of the antibody to the protein on the outer mitochondrial membrane. H89 enhanced the effect of the anti-CNPase antibody, while CmZ abolished the effect of the anti-CNPase antibody upon the mPTP opening. The anti-CNPase antibody in the presence/absence of CmZ and H89 upon the opening of mPTP in any event changed the level of pAkt and pGSK3β, which suggests the involvement of CNPase in the regulation of stress kinases. In the absence of anti-CNPase antibody we observed of correlation between pAkt/Akt and pGSK3β/GSK3β ratios. On the other hand, correlation between pAkt/Akt and pGSK3β/GSK3β ratios was broken when anti-CNPase antibody was added to mitochondria suspension. We suggest that CNPase may be involved in the regulation of these kinases. When the pore opened, the level of MEL decreased, whereas the level of CNPase remained unchanged. If the addition of CmZ to anti-CNPase antibody-loaded RBM did not alter the level of MEL, the level of CNPase increased, whereas H89 had an opposite effect on the levels of MEL and CNPase under these conditions. These results suggest the involvement of mitochondrial CNPase and MEL in the signal transduction upon the functioning of mPTP. The mechanism by which this can occur needs to be established.

## 4. Materials and Methods

### 4.1. Animals

Mitochondria were isolated from the total brain of two-month-old male Wistar rats. All experiments were performed in accordance with the “Regulations for Studies with Experimental Animals” (Decree of the Russian Ministry of Health of 12 August 1997, No. 755). The protocol was approved by the Commission on Biological Safety and Ethics of the Institute of Theoretical and Experimental Biophysics, Russian Academy of Science (November 2014, protocol N45). A total of six male rats were used in the experiments.

### 4.2. Isolation of Rat Brain Mitochondria

Rat brains were rapidly removed (within 30 s) and placed in a solution containing 320 mM sucrose, 0.5 mM EDTA, 0.5 mM EGTA, 0.02% bovine serum albumin (fraction V, fatty acid free) and 10 mM Tris-HCl, pH 7.4. All solutions were used ice-cold; all manipulations were carried out at +4 °C. The brain tissue was homogenized in a glass homogenizer; the ratio of brain tissue to isolation medium was 1:10 (*w*/*v*). The homogenate was centrifuged at 2000× *g* for 3 min. The crude mitochondrial pellet was obtained by centrifugation of the 2000× *g* supernatant at 12,500× *g* for 10 min. Then, mitochondria were purified in a discontinuous Percoll gradient (3%–10%–15%–24%). Non-synaptic RBM were suspended in an ice-cold solution containing 320 mM sucrose and 10 mM Tris-HCl, pH 7.4, additionally washed by centrifugation at 11,500× *g* for 10 min, and resuspended in the same buffer. The protein concentration was determined by the Bradford method (Bio-Rad Protein assay; Bio-Rad, Munich, Germany) using bovine serum albumin as standard and was 25–30 mg/mL in the stock mitochondrial suspension. The integrity of non-synaptic mitochondria was investigated in some experiments by electron microscopy [64].

### 4.3. Evaluation of Mitochondrial Functions

The Ca^2+^ retention capacity was determined with a Ca^2+^-sensitive electrode (Nico, Russia), and the oxygen consumption rate was measured with a Clark-type O_2_ electrode that was integrated into a 1-mL multifunctional chamber [65] for the simultaneous registration of Ca^2+^ flux and respiratory activity. Mitochondria (1 mg protein/mL) were incubated in a medium containing 125 mM KCl 10 mM Tris-HCl, 0.4 mM K_2_HPO_4_, and 5 µM rotenone (inhibitor of mitochondrial electron transport—I Complex), pH 7.4, at 25 °C. Succinate (5 mM potassium succinate) was used as a mitochondrial respiratory substrate. PTP opening in non-synaptic RBM was induced by a threshold Ca^2+^ load (the first addition of Ca^2+^ contained 50 nmoles per mg protein, the subsequent addition of Ca^2+^ was 100 nmoles per mg of protein). All experiments were performed in an opened chamber.

The swelling of non-synaptic RBM was determined by measuring changes in light scattering of the mitochondrial suspension at 540 nm (A540) using a Tecan I-Control infinite 200 spectrophotometer at 25 °C. The standard incubation conditions for the swelling assay were 125 mM KCl, 10 mM Tris, 0.4 mM KH_2_PO_4_, 5 mM succinate, and 5 µM rotenone. Swelling was initiated by the addition of 450 nmol of Ca^2+^/mg protein. The concentration of protein in a well was 0.5 mg protein/mL. The swelling process was characterized by the time needed to reach the half-maximal light scattering signal (T1/2). The top and the bottom plateau for half-maximal light scattering signal (A/2) was assigned for each curve individually using SigmaPlot functions.

### 4.4. Sample Preparation

Aliquots of a mitochondrial suspension (100 μL) taken from the chamber were placed in an Eppendorf tube. For protein phosphorylation, the mitochondrial suspension was supplemented with unlabeled Mg^2+^/ATP to achieve physiological concentrations of 2 mM Mg^2+^ and 400 µM ATP. Samples were incubated for 3 min in the presence of 1.5 µM oligomycin. The reaction was stopped by the addition of solubilization 4× Laemmli sample buffer (20 μL) and heated in a boiling water bath for 3 min. Twenty micrograms of mitochondrial samples was applied to the gel and subjected to electrophoresis followed by Western blot analysis.

### 4.5. Electrophoresis and Immunoblotting of Mitochondrial Proteins

The samples obtained were separated under denaturing conditions by 12.5% SDS-PAGE and transferred to a nitrocellulose membrane. Precision Plus Pre-stained Standards from Bio-Rad Laboratories (Hercules, CA, USA) were used as markers. After overnight blocking, the membrane was incubated with the appropriate primary antibody. The monoclonal anti-CNPase antibody (anti-CNPase Ab) was obtained as described [66] (dilution 1:10,000); polyclonal rabbit phospho-Akt (Cat. #4058) (Ser473), Akt (Cat. #9272), and phospho-GSK3β (Ser9) (Cat. #9356) antibody were from Cell Signalling (dilution 1:500); the monoclonal GSK3β antibody was from Invitrogen (Cat. #AHO1302, dilution 1:500); and the polyclonal anti-melatonin receptor antibody was from Abcam (Cat. #ab128664, dilution 1:1000). The monoclonal COX IV antibody (Abcam, Cat. #ab14744) was used as a loading control. Immunoreactivity was determined using the appropriate secondary antibody conjugated to horseradish peroxidase (Jackson Immuno Research, West Grove, PA, USA). Peroxidase activity was detected with ECL chemiluminescence reagents (Pierce, Rockford, IL, USA).

### 4.6. Statistical Analysis

For statistical analysis, the relative levels of protein density were expressed as mean ± SD from at least three independent experiments. The statistical significance of the difference between the mean values was evaluated using the Student’s *t*-test. The difference was considered significant at *p* < 0.05.

## Figures and Tables

**Figure 1 ijms-19-03499-f001:**
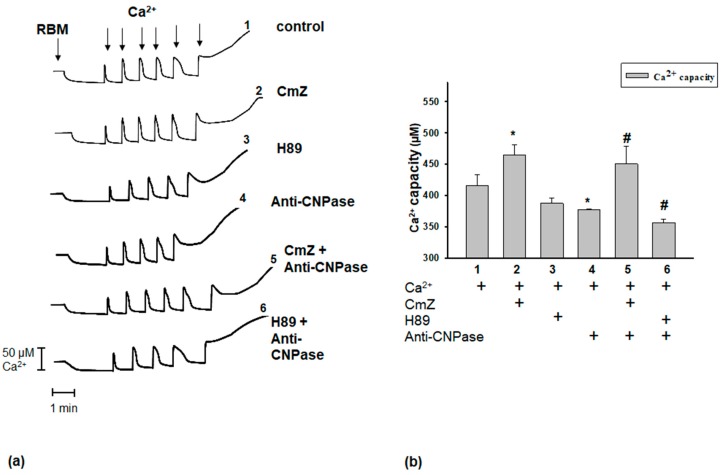
Effect of calmidozolium (CmZ), H89, anti-CNPase antibody, and their combined effect on Ca^2+^ capacity in rat brain mitochondria (RBM). Isolated RBM (1.0 mg/mL protein) were incubated in an electrode chamber under conditions described in Materials and methods. (**a**) Representative traces of Ca^2+^ fluxes in RBM. Arrows show the times at which CaCl_2_ was applied (the first addition, 50 nmol of Ca^2+^ per mg of protein; the subsequent addition, 100 nmol each of Ca^2+^ per mg of protein). (**b**) Quantitative analysis of Ca^2+^-capacity in the presence/absence of CmZ (30 μM), H89 (5 μM), and anti-CNPase antibody (0.18 µg/mL). The values shown are the means ± SD from three independent experiments; * *p* ≤ 0.05 vs. the Ca^2+^ retention capacity without any additions, # *p* ≤ 0.05 compared with the Ca^2+^-capacity in the presence of anti-CNPase antibody. CNPase: 2′,3′-cyclic nucleotide-3′-phosphodiesterase.

**Figure 2 ijms-19-03499-f002:**
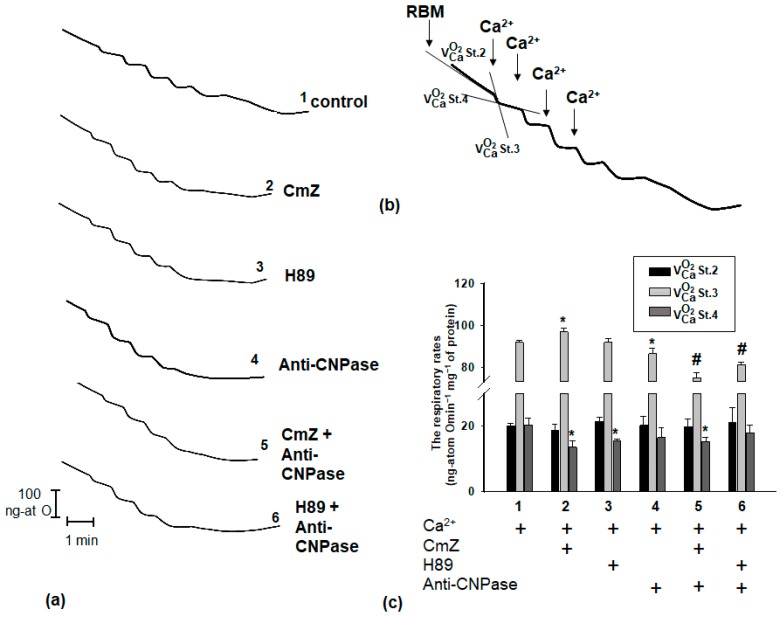
Effect of CmZ, H89, and anti-CNPase antibody, and their combined effect on respiratory activity in RBM. The conditions were the same as in Figure 1. (**a**) Representative curves of respiratory activities. (**b**) A schema of calculation of oxygen consumption in states V^O2^_Ca_ St.2, V^O2^_Ca_ St. 3, V^O2^_Ca_ St. 4. (**c**) Quantitative analysis of respiratory activity in the presence/absence of CmZ (30 μM), H89 (5 μM), and anti-CNPase antibody (0.18 µg/mL). The values shown are the means ± SD from three independent experiments; * *p* ≤ 0.05 versus respiratory activity without any additions, # *p* ≤ 0.05 versus respiratory activity in the presence of anti-CNPase antibody.

**Figure 3 ijms-19-03499-f003:**
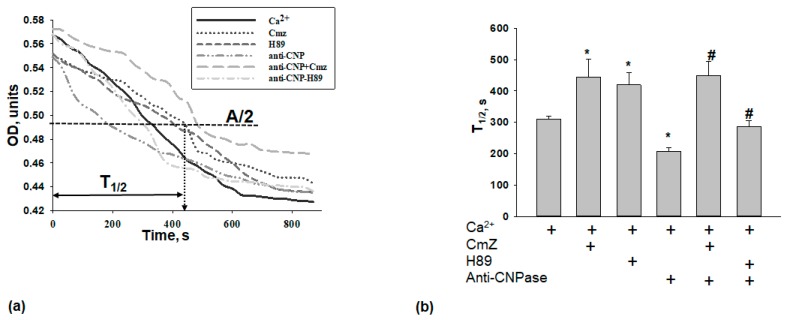
Effect of CmZ, H89, and anti-CNPase antibody, and their combined effect on RBM swelling. (**a**) Curves of RBM swelling in the presence of CmZ (30 μM), H89 (5 μM), and anti-CNPase antibody (0.18 µg/mL); (**b**) Average results of the half-time (T_1/2_) swelling. The values shown are the means ± SD from three independent experiments; * *p* ≤ 0.05 vs. T_1/2_ without any additions, # *p* ≤ 0.05 vs. T_1/2_ in the presence of anti-CNPase antibody. OD—optical density, A/2—half amplitude of mitochondrial swelling.

**Figure 4 ijms-19-03499-f004:**
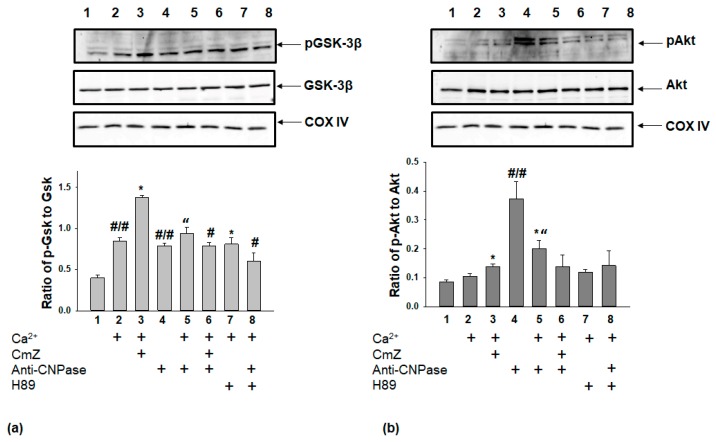
The phosphorylation states of stress-activated protein kinases (GSK3β and Akt) in RBM. (**a**) The ratio of pGSK3β to total GSK3β. (**b**) The ratio of pAkt to total Akt. Upper parts represent Western blots of a mitochondrial suspension after the isolation of RBM (see Section 4). Membranes were stained with antibody specific to pGSK3β, GSK3β, pAkt, and Akt. Lower parts—quantitation of immunostaining using computer-assisted densitometry. Bar graphs represent the ratios of pGSK3β to total GSK3β (**a**) and of pAkt to total Akt (**b**) in RBM after the addition of CmZ (30 μM), H89 (5 μM), and anti-CNPase antibody (0.18 µg/mL). The protein band intensity was quantified after normalization with respect to Cytochrome c oxidase subunit IV (COX IV). The values shown are the means ± SD from three independent experiments; #/# *p* ≤ 0.05 vs. the pGSK3β/GSK3β ratio without any additions, * *p* ≤ 0.05 vs. pGSK3β/GSK3β ratio in the presence of Ca^2+^, “*p* ≤ 0.05 vs. pGSK3β/GSK3β ratio in the presence of anti-CNPase antibody, # *p*—versus pGSK3β/GSK3β ratio in the presence of anti-CNPase antibody and Ca^2+^.

**Figure 5 ijms-19-03499-f005:**
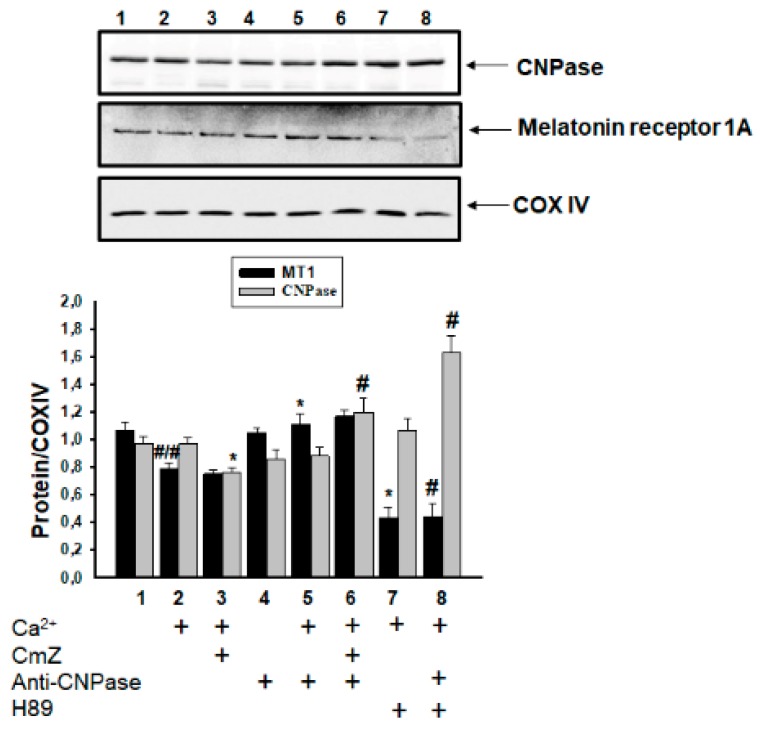
Effect of CmZ, H89, and anti-CNPase antibody, and their combined effect on the level of MT1 and CNPase in RBM. Upper part—western blot of a mitochondrial suspension after the isolation of RBM (see Materials and methods). Membranes were stained with antibody specific to Melatonin receptor A1 (MT1) and CNPase. Lower part—quantitation of immunostaining using computer-assisted densitometry. Bar graphs represent the immunoreactivity of MEL MT1 receptor and CNPase in RBM after the addition of CmZ (30 μM), H89 (5 μM), and anti-CNPase antibody (0.18 µg/mL). The protein band intensity was quantified after normalization with respect to COX IV. The values shown are the means ± SD from three independent experiments; #/# *p* ≤ 0.05 vs. MT1 level in in control conditions without any additions, * *p* ≤ 0.05 vs. the MT1 and CNPase levels in the presence of Ca^2+^. # *p* ≤ 0.05 vs. MT1 and CNPase levels in the presence of anti-CNPase antibody and Ca^2+^.

**Figure 6 ijms-19-03499-f006:**
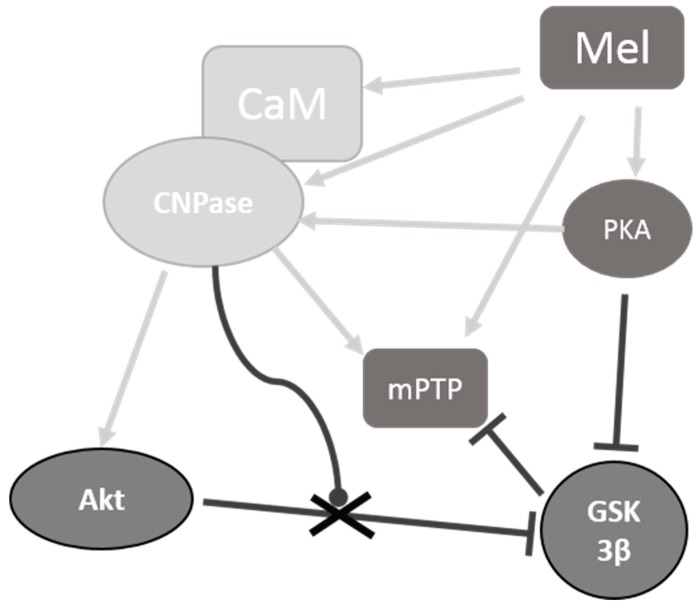
A hypothetical scheme of the involvement of melatonin (MEL) and CNPase in the regulation of mitochondrial permeability transition pore (mPTP) functioning. The figure shows possible protein phosphorylation-mediated mechanisms in rat brain mitochondria. According to our concept, CNPase, which is a target of the action of MEL, is capable to participate in the regulation of signaling pathways in mitochondria by changing the level of phosphorylation. In particular, CNPase and MEL are involved in the regulation of mPTP thereby inducing its opening. The regulation of mPTP opening is mediated through the phosphorylation by protein kinase A (PKA) and implies the participation of stress kinases (AKT and GSK 3β). CNP and MEL are also capable of binding CaM, indicating their involvement in calcium signaling and cAMP signaling pathways.

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
