# Peer review of "Possible Involvement of 2′,3′-Cyclic Nucleotide-3′-Phosphodiesterase in the Protein Phosphorylation-Mediated Regulation of the Permeability Transition Pore"

_ijms, 2018, doi:10.3390/ijms19113499_

Round 1
Reviewer 1 Report
Dear Authors,
Below you can find the reviewer`s suggestions.
line 1 - I suggest: "...regulation of the mitochondrial permeability pore functioning",
line 14 - it sounds like CaM act as a protein kinase - please rewrite,
line 39 - not "causes a phosphorylation" but "catalyses a phosphorylation and inactivates",
line 84 - protein kinase-alpha???
line 296 - ..
References - there are only 3/63 positions from the years 2016-2017, there are no reference from 2018.
Author Response
We cordially thank Reviewer for attentive reading of our manuscript and for their time.
Response to Reviewer 1 Comments
Dear Authors,
Below you can find the reviewer`s suggestions.
line 1 - I suggest: "...regulation of the mitochondrial permeability pore functioning",
line 14 - it sounds like CaM act as a protein kinase - please rewrite,
line 39 - not "causes a phosphorylation" but "catalyses a phosphorylation and inactivates",
line 84 - protein kinase-alpha???
line 296 - ..
References - there are only 3/63 positions from the years 2016-2017, there are no reference from 2018.
Response : We have taken all your remarks and recommendations.
Reviewer 2 Report
The manuscript by Baburina, et al. explores interplay of Ca2+-related signalling and phosphorylation-mediated pathways in isolated rat brain mitochondria (RBM). While the authors have performed a variety of experiments, I would recommend a major revision of the given paper not from the experimental, but rather from the representational aspect. Currently, the manuscript as a whole lacks focus, and its several parts (including abstract, introduction and discussion) are poorly structured. Although authors provide a sufficient number of references to the previous works of their own and others, it is extremely difficult to follow why the study of the chosen pathways was undertaken, and what were the initial hypotheses prior to generating large amount of the experimental data.
I would ask authors to specifically address the following issues:
1. Please include a schematic illustration of all the studied pathways (i.e., with arrows indicating upstream and downstream signalling and activation/inactivation processes) according to your current understanding. It would immensely aid the reader in getting though the manuscript.
2. From the results and discussion, it seems that authors are confused about the structure of melatonin. It is a small molecule which derived from tryptophan, so it is not correct to refer to its expression (see lane 302) and I have serious doubts that it can be visualized on Western blot (Figure 5).
3. There is no consistency between the results obtained with different methods (Figures 1-5). Why does anti-CNPase sometimes have the same effect as CmZ in the presence of Ca2+ and sometimes the opposite one (relative to Ca2+-only control)? The same applies to H89 versus Cmz or H89 versus anti-CNPase in the presence of Ca2+. This is the part where thorough discussion is needed.
4. One of the conclusions mentioned in the abstract is as follows: “…phospho-Akt and phospho-GSK3β are involved in the regulation of CNPase function, which in turn plays an important role in mPTP functioning” (lanes 26-27). How can such a conclusion be drawn, if according to data, manipulation of RBM with anti-CNPase resulted in alteration of levels of phosphorylated proteins, not vice versa?
5. Why is there no correlation between pGSKβ/GSK ratio and pAkt/Akt ratio, taken that Akt phosphorylates GSK?
There are also some minor issues that require attention:
A. There are several inconsistencies throughout the text of the manuscript, in addition to those outlined above. In the abstract, the authors state that “Calmodulin is capable to regulate target proteins by phosphorylation” (lane 14), which is wrong, as CaM is not a kinase. The “phosphor-inactivation” in lane 285 should probably read as ‘phospho’ or ‘phosphoryl’. I am also quite sure that at the end of the topic of the manuscript, the word ‘pore’ is missing.
B. The Western blot membranes should be provided in full in supplementary, together with indication of the molecular weight of ladder or immunodetected bands in kDa.
C. The information on the antibodies used should include catalogue numbers of the antibodies. Have the authors produced themselves the anti-CNPase antibody, or have they obtained it from the authors of ref #63?
D. The authors claim that according to literature, “The addition of the anti-CNPase antibody to a mitochondrial suspension was found to protect mitochondria from mPTP opening” (lanes 71-72). Why does anti-CNPase have an opposite effect in the current work?
E. The applied concentration of Ca2+ should also be reported in the molarity units, not only nmol per mg of total protein.
Author Response
We cordially thank Reviewer for attentive reading of our manuscript and for their time.
Response to Reviewer 2 Comments
Response 1: The scheme was added to the manuscript.
Response 2: The physiological actions of melatonin are mediated by two G-protein coupled membrane receptors, MT1 (Reppert SM, Weaver DR, Ebisawa T. Cloning and characterization of a mammalian melatonin receptor that mediates reproductive and circadian responses. Neuron. 1994;13:1177–85) and MT2 (Reppert SM, Godson C, Mahle CD, Weaver DR, Slaugenhaupt SA, Gusella JF. Molecular characterization of a second melatonin receptor expressed in human retina and brain: the Mel1b melatonin receptor. Proc Natl Acad Sci USA. 1995;92:8734–8) and the MT3 binding site (Nosjean O, Ferro M, Coge F, et al. Identification of the melatonin-binding site MT3 as the quinone reductase 2. J Biol Chem. 2000;275:31311–7), which belongs to the family of the quinone reductases). In mammals, there are two isoforms of the melatonin receptor, MT1 and MT2. MT1 and MT2 are found in the brain and peripheral tissues (Coto-Montes, A.; Tomas-Zapico, C.; Escames, G.; Leon, J.; Rodriguez-Colunga, M. J.; Tolivia, D.; Acuna-Castroviejo, D., Specific binding of melatonin to purified cell nuclei from mammary gland of swiss mice: day-night variations and effect of continuous light. J Pineal Res 2003, 34, (4), 297-301.
Drew, J. E.; Barrett, P.; Mercer, J. G.; Moar, K. M.; Canet, E.; Delagrange, P.; Morgan, P. J., Localization of the melatonin-related receptor in the rodent brain and peripheral tissues. J Neuroendocrinol 2001, 13, (5), 453-8.
Pozo, D.; Garcia-Maurino, S.; Guerrero, J. M.; Calvo, J. R., mRNA expression of nuclear receptor RZR/RORalpha, melatonin membrane receptor MT, and hydroxindole-O-methyltransferase in different populations of human immune cells. J Pineal Res 2004, 37, (1), 48-54.). It was found that both MT1 and MT2 are expressed in brain mitochondria, suggesting that the mitochondrial cell-death pathways might be mediated in part by interactions of MEL with its receptors, although it is possible that MEL has additional, non-receptor-mediated targets.
In our experiments, we used the anti-melatonin related receptor antibody.
Response 3: Our results showed that the addition of the anti-CNPase antibody accelerated the mPTP opening, whereas CmZ inhibited this process. CmZ in combination with the antibody eliminated the effect of the anti-CNPase antibody, probably due to the disturbance of the CNPase-СаМ interaction. We observed a similar effect when studying mitochondrial swelling under similar conditions. The possible breakage of bonds may lead to an increase in the level of CNPase, and an increased level of MEL retains CNPase in mitochondria. The content of рAkt and рGSK3β did not change, which just led to the inhibition of мРТР opening upon the combined action of CmZ and the anti-CNPase antibody.
Н89 inhibits PKA, and the decrease in the content of MEL also diminishes the effect of PKA. The level of CNPase increases to protect mitochondria against damage. MEL inhibits рAkt, and its level decreases. рAkt in turn inhibits рGSK3β, and the induction of mРТР is enhanced.
Response 4: We have shown earlier that the level of CNPase phosphorylation changed when the pore was open; later we have found that changes in the level of CNPase phosphorylation occur by the action of РКА and РКС (Krestinina, O. V.; Odinokova, I.; Baburina, Y. L.; Azarashvili, T. S., Detection of Protein Kinase A and C Target Proteins in Rat Brain Mitochondria. Biol Membrany 2017, 34, (5), 42-47.); i.e., these protein kinases are involved in CNPase phosphorylation. The changes in the level of pAkt and pGSK3β in the presence of the CNPase antibody suggests that CNPase may be a target of these kinases.
Response 5: Activated Akt is localized to diverse subcellular compartments: these include the Golgi, endoplasmic reticulum, and the nucleus. However Akt could be localized in mitochondria. Akt also has a direct effect in mitochondria, which is mediated by phosphorylation of hexokinase-II, resulting in protection of the mitochondria from oxidant or Ca2+-induced mitochondrial permeability transition pore (mPTP) opening. GSK can be localized in cytosol and nucleus. However, it was revealed that GSK3β is present in mitochondria in rat cerebellum. The mechanism of GSK3β’s translocation to the mitochondria is unclear. For example, «there are contradictory findings that GSK3β mediates tumor promotion and/or GSK3β shows anti-proliferative effects in certain types of tumors including colon and pancreatic cancer».
We believe that the processes we examined proceed by different mechanisms, which probably is related to different pathways of Akt and GSK transport in mitochondria; therefore, a goal of our further studies will be the elucidation of these mechanisms.
There are also some minor issues that require attention:
A. There are several inconsistencies throughout the text of the manuscript, in addition to those outlined above. In the abstract, the authors state that “Calmodulin is capable to regulate target proteins by phosphorylation” (lane 14), which is wrong, as CaM is not a kinase. The “phosphor-inactivation” in lane 285 should probably read as ‘phospho’ or ‘phosphoryl’. I am also quite sure that at the end of the topic of the manuscript, the word ‘pore’ is missing.
We made proper corrections.
B. The Western blot membranes should be provided in full in supplementary, together with indication of the molecular weight of ladder or immunodetected bands in kDa.
We added full Western blots to the supplementary files.
C. The information on the antibodies used should include catalogue numbers of the antibodies. Have the authors produced themselves the anti-CNPase antibody, or have they obtained it from the authors of ref #63?
The information concerning the antibodies was added to the Materials and Methods section.
D. The authors claim that according to literature, “The addition of the anti-CNPase antibody to a mitochondrial suspension, which accelerates the mPTP opening, confirms our recent observation that CNPase may be involved in the mPTP functioning” (lanes 71-72). Why does anti-CNPase have an opposite effect in the current work?
We corrected this sentence. Now it reads as follows: “The addition of the anti-CNPase antibody to a mitochondrial suspension, which accelerates the mPTP opening, confirms our recent observation that CNPase may be involved in the mPTP functioning” (lanes 71-72).
E. The applied concentration of Ca2+ should also be reported in the molarity units, not only nmol per mg of total protein.
In the text, the threshold concentration of Са2+ the molarity units under different conditions were indicated. In the legends, we indicated what amount of Са2+ was in each additive.
Round 2
Reviewer 2 Report
The revision of the manuscript carried out by authors has improved its quality, yet there are still several major issues.
First of all, if authors used antibody to melatonin receptor, not melatonin itself (as is evident from their answers to review), then they should rewrite all sentences in the text referring to the current work (as well as Figure 6) using MT1 instead of MEL accordingly, and change ‘melatonin’ in Figure 5 to ‘melatonin receptor 1A’.
Second, the authors’ comment on CNPase and Akt/GSK3b pathways (“The changes in the level of pAkt and pGSK3β in the presence of the CNPase antibody suggests that CNPase may be a target of these kinases” – answers to reviewer) clearly shows misunderstanding of principles of the upstream and downstream signalling. If there is alteration in pAkt levels after anti-CNPase treatment, it indicates that Akt is the target of CNPase, not vice versa.
Third, authors give self-contradicting explanations to the question on the absent correlation between pAkt/Akt and pGSK3b/GSK3b ratios. On one hand, they state that the absence of correlation might be caused by differences in mitochondrial transport of the kinases (“the processes we examined proceed by different mechanisms, which probably is related to different pathways of Akt and GSK transport in mitochondria” – answers to reviewer), yet on the other hand, they emphasize presence of signalling relationship between those (“рAkt in turn inhibits рGSK3β, and the induction of mРТР is enhanced” – answers to reviewer, discussion part of the manuscript and Figure 6). Please develop a clear concept considering this topic and present it throughout the manuscript.
Finally, the authors should comment in the experimental section on how the half-lives were established in Figure 3b. Were the curves in Figure 3a fitted to some mathematical function, or was T1/2 just manually established (in the latter case, how was the top and the bottom plateau assigned, especially in case of curves anti-CNPase+H89 and anti-CNPase+Cmz)?
Author Response
We cordially thank Reviewer for attentive reading of our manuscript and for their time.
1. We have taken this remark into consideration.
2. The changes in the level of pAkt and pGSK3β in the presence of the anti-CNPase antibody suggests that CNPase may be involved in the regulation of these kinases.
3. In the absence of anti-CNPase antibody we observed of correlation between pAkt/Akt and pGSK3β/GSK3β ratios. On the other hand, correlation between pAkt/Akt and pGSK3β/GSK3β ratios is broken when anti-CNPase antibody was added to mitochondria suspension. It has probably affected Akt and GSK transport in mitochondria. Therefore, we suggests that CNPase may be involved in the regulation of these kinases.
4. The information concerning the how the half-lives were established was added to the Materials and Methods section.
Round 3
Reviewer 2 Report
I would like to thank authors for their will to cooperate, and consider the current version of the manuscript acceptable for publication